# OTKGE: Multi-modal Knowledge Graph Embeddings via Optimal Transport

**Zongsheng Cao**[1,2]    **Qianqian Xu**[3*]    **Zhiyong Yang**[4]
**Yuan He**[5]    **Xiaochun Cao**[6,1]    **Qingming Huang**[4,3,7,8*]

[1] SKLOIS, Institute of Information Engineering, CAS
[2] School of Cyber Security, University of Chinese Academy of Sciences
[3] Key Lab. of Intelligent Information Processing, Institute of Computing Tech., CAS
[4] School of Computer Science and Tech., University of Chinese Academy of Sciences
[5] Alibaba Group
[6] School of Cyber Science and Tech., Shenzhen Campus, Sun Yat-sen University
[7] BDKM, University of Chinese Academy of Sciences
[8] Peng Cheng Laboratory
caozongsheng@iie.ac.cn    xuqianqian@ict.ac.cn
yangzhiyong21@ucas.ac.cn    heyuan.hy@alibaba-inc.com
caoxiaochun@mail.sysu.edu.cn    qmhuang@ucas.ac.cn

## Abstract

Multi-modal knowledge graph embeddings (KGE) have caught more and more attention in learning representations of entities and relations for link prediction tasks. Different from previous uni-modal KGE approaches, multi-modal KGE can leverage expressive knowledge from a wealth of modalities (image, text, *etc.*), leading to more comprehensive representations of real-world entities. However, the critical challenge along this course lies in that the multi-modal embedding spaces are usually heterogeneous. In this sense, direct fusion will destroy the inherent spatial structure of different modal embeddings. To overcome this challenge, we revisit multi-modal KGE from a distributional alignment perspective and propose *optimal transport knowledge graph embeddings* (**OTKGE**). Specifically, we model the multi-modal fusion procedure as a transport plan moving different modal embeddings to a unified space by minimizing the Wasserstein distance between multi-modal distributions. Theoretically, we show that by minimizing the Wasserstein distance between the individual modalities and the unified embedding space, the final results are guaranteed to maintain consistency and comprehensiveness. Moreover, experimental results on well-established multi-modal knowledge graph completion benchmarks show that our OTKGE achieves state-of-the-art performance.

## 1 Introduction

In the past decades, with the emergence and development of numerous knowledge graphs (KGs) [1, 2], a spectrum of related applications has been widely facilitated, *e.g.*, question answering [3, 4], semantic search [5, 6], and recommendation systems [7, 8]. Due to the incompleteness of real-world knowledge graphs, link prediction becomes an important procedure for constructing knowledge graphs. To achieve this goal, Knowledge Graph Embedding (KGE) [9, 10, 11, 12, 13] attracts more and more attention. It can learn the low-dimensional representations of entities and relations and thus predict missing links.

Nowadays, most of the KGE methods [11, 12, 13] mainly focus on the uni-modal knowledge graphs represented with pure symbols or concepts. However, most data sources we can access exhibit a

---

*Corresponding authors.

36th Conference on Neural Information Processing Systems (NeurIPS 2022).

multi-modal property, where information from text, image, and video coexists and comprehensively describes the underlying target. In recent years, researchers [14, 15, 16, 17] point out that knowledge extracted from such multi-modal data can significantly improve the quality of the learned knowledge graph representations. To that end, the study of the multi-modal KGE raises a new wave, and a great many meaningful work [18, 16, 19, 15, 20] emerges and achieves remarkable success. Specifically, they usually embed the multi-modal knowledge (*e.g.,* texts, pictures) of the entity into different spaces, then fuse these multi-modal embeddings with the help of operations such as `concat` or `mean`. The resulting embedding is ultimately employed as a unified representation for the multi-modal entity. However, since the embeddings of different modalities are usually in various heterogeneous spaces, direct fusion might destroy the intrinsic distribution and thus lead to inconsistent and incomprehensive representation in the unified space. Based on this fact, this paper focuses on the following question:

> *Can we find a new KGE model, which is capable of learning unified representations of multi-modal entities while overcoming the problem of spatial heterogeneity?*

To address this issue, we propose a novel multi-modal KGE method called *Optimal Transport Knowledge Graph Embeddings* (**OTKGE**). Specifically, we formulate the fusion procedure of multi-modal knowledge as an optimal transport (OT) problem [21]. As shown in Figure 1, OT can move different modal embeddings to a unified aligned space by an optimal transport plan while overcoming spatial heterogeneity by minimizing the Wasserstein distance between different distributions. In this way, we can facilitate the information interaction between multi-modal embeddings and obtain a unified representation by the Wasserstein barycenter. On top of this, we theoretically demonstrate that the divergence between the distribution of individual source modality and that of the unified space can be bounded by the Wasserstein distance, suggesting that our proposed method can overcome the spatial heterogeneity issue. Moreover, experimental results demonstrate the superiority of OTKGE on multi-modal knowledge graph completion tasks.

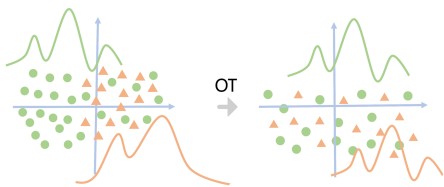

Figure 1: An illustration of the optimal transport (OT) for multi-modal entities. Green and red colors represent the different modalities. Lines represent distributions of different modal entities while circles and triangles represent the embeddings of different modal entities.

In a nutshell, our contributions are summarized as three-fold:

- For the modal fusion problem in multi-modal KGE, we propose a new fusion approach based on optimal transport, which can effectively tackle the modal spatial heterogeneity by reducing the Wasserstein distance between different modal distributions.

- Theoretically, we provide an upper bound for the target errors from source multi-modal spaces to the unified space in OT fusion. It can provide a theoretical guarantee for modal fusion while overcoming spatial heterogeneity.

- Extensive experiments show that OTKGE achieves state-of-the-art performance on two multi-modal knowledge graph completion tasks.

## 2 Related Work

The study of knowledge graphs has attracted widespread attention in recent years. According to the number of used modalities, the previous methods mainly fall into two branches including uni-modal KGEs and multi-modal KGEs.

### 2.1 Uni-modal Knowledge Greaph Embeddings

In the past decades, researchers have proposed various embedding approaches to achieve uni-modal KGE, named translational distance models, semantic matching models, and neural network models. First of all, translational distance models are derived from the popular method TransE [9], which proposes to model the relation as the translation operation between entities, and adopts the principle $head\ entity + relation = tail\ entity$. Subsequently, there are many variants of TransE, *e.g.*, TransH [10], TransR [22] and TransD [23], which model the relation as the projection operation and model

the entity as the vectors, respectively. Despite the practical advantages, they cannot model some relation patterns (*e.g.,* symmetric, inverse, and composition) [12].

Semantic matching approaches aim to capture and match latent semantics between entities, which models the plausibility of the triple by semantic matching. For instance, some representative methods are RESCAL [24], DistMult [25], ComplEx [11], and HolE [26]. It is worth mentioning that RotatE [12] models the relation as the rotation in two-dimensional space, thus realizing the goal to model symmetry/antisymmetry, inversion, and composition patterns. Inspired by RotatE, there are also some variants of RotatE, such as QuatE [13], DualE [27], and ROTH [28], which introduce rotations into other geometric spaces.

In addition to the conventional methods, there are also some models using neural networks to produce KGE with remarkable effects, such as R-GCN [29], ConvE [30], ConvKB [31], KBGAT [32], and A2N [33]. However, some of them are usually computationally expensive and may require pre-train embeddings.

### 2.2 Multi-modal Knowledge Graph Embeddings

Beyond uni-modal KGE methods, recently a great many multi-modal KGE methods have been proposed, which can be divided into neural network methods and translation methods.

Neural network methods utilize neural networks to learn embeddings for structural and multi-modal knowledge. For instance, MMRFAN [17] and MKRL [14] adopt the unified space to learn the textual knowledge and visual knowledge of entities and thus realize the multi-modal KGE. Unfortunately, these models such as MMRFAN are designed for some specific cases (*e.g.,* medical knowledge graphs), and they may be not suitable for other general graph structures [20].

Translation methods follow the principle of TransE and model the relation as the translation for multi-modal entities. For instance, DKRL [18] learns the representations of KGs by exploiting the textual and structural information while IKRL [16] learns representations by integrating visual information and structured information. However, both two models can only use one of the textual information and visual information. To tackle this problem, [19] proposes to take the sum of sub-energy functions as the overall score function, which aims to utilize the textual information, visual information, and structured KGs simultaneously. Then TransAE [15] adds a multi-modal autoencoder based on TransE, which can achieve the goal to obtain the embeddings of multi-modal KGE. Afterward, MMKRL [20] integrates structured knowledge and multi-modal knowledge in a unified space with the assistance of concat fusion operation. In a word, these models fuse heterogeneous multi-modal spaces by concat or mean operations, neglecting the differences between distributions. Therefore, learning embeddings for multi-modal knowledge graphs is still an open problem to be tackled.

## 3 Background on Optimal Transport

Before giving an overview of the optimal transport (OT) problem, we first introduce its definition and present a short introduction of the related background. Specifically, OT serves as a way to compare two probability distributions, which can provide a transport plan to transfer one point to another.

**Optimal Transport Problem [21].** Suppose we have two set of points $X = (x^{(1)}, \cdots, x^{(n)}) \in \mathbb{R}^n$ and $Y = (y^{(1)}, \cdots, y^{(m)}) \in \mathbb{R}^m$ with the Dirac (unit mass) distribution denoted as $\delta(\cdot)$. Our first goal is to obtain the empirical probability measures $\mu$ and $\nu$ for $X$ and $Y$, respectively. To achieve this, we let $\mu = \sum_{i=1}^{n} \alpha_i \delta(x^{(i)})$ and $\nu = \sum_{i=1}^{m} \beta_i \delta(y^{(i)})$, where the weight $\boldsymbol{\alpha} = (\alpha_1, \cdots, \alpha_n)$ and $\boldsymbol{\beta} = (\beta_1, \cdots, \beta_m)$ can be regarded as the probability simplexes. Subsequently, we design a ground cost to measure the distance from point $x^{(i)}$ to $y^{(j)}$, which is denoted as $C_{ij}$. Further, we need to calculate the transport coupling $T$ between $\mu$ and $\nu$, and the $T_{ij}$ can measure the joint probability observing $x^{(i)}$ and $y^{(j)}$. Conveniently, it can be obtained in the procedure of solving the following: $\mathbf{OT}(\mu, \nu, C) := \min_T \langle T, C \rangle$, where $\{T \in \mathbb{R}_+^{(n \times m)} | T\mathbf{1}_m = \boldsymbol{\alpha}, T^T \mathbf{1}_n = \boldsymbol{\beta}\}$. It is worth noting that $\langle T, C \rangle := tr(T^T C)$ is formulated as the Frobenius inner product of matrices.

**Wasserstein Distance.** If we formulate the cost function with the p-norm, *i.e.*, $C_{ij} = ||x^{(i)}, y^{(j)}||_p^p$, then the $p$-th Wasserstein distance between two empirical probabilistic measure can be defined as

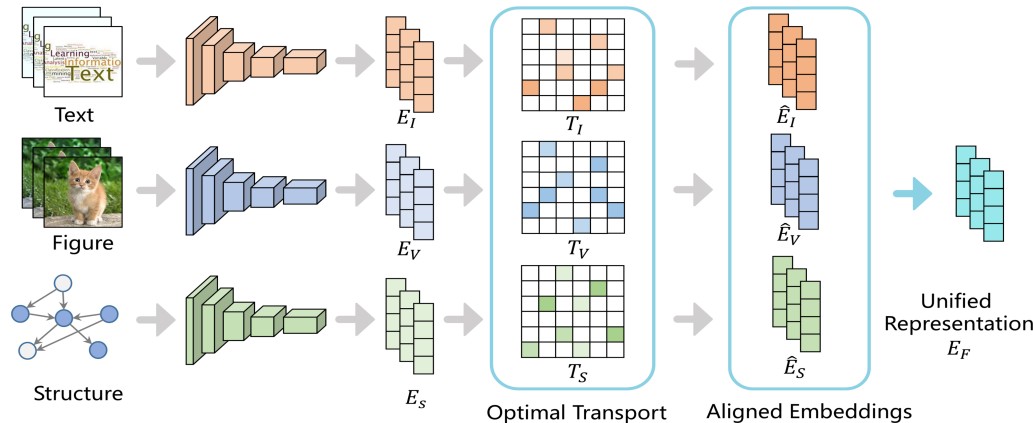

Figure 2: The overall framework for OT fusion. $\boldsymbol{E}_S, \boldsymbol{E}_I, \boldsymbol{E}_V, \boldsymbol{E}_F$ are structural, linguistic, visual, and unified embeddings, respectively. $\boldsymbol{T}_I, \boldsymbol{T}_S, \boldsymbol{T}_V$ are optimal transport plans. $\hat{\boldsymbol{E}}_S, \hat{\boldsymbol{E}}_I, \hat{\boldsymbol{E}}_V$ are the according aligned embeddings, respectively. In our method, $\boldsymbol{T}_S$ is an identity matrix and $\hat{\boldsymbol{E}}_S = \boldsymbol{E}_S$ since the multi-modal knowledge is to align with the structural knowledge.

$\mathcal{W}_p(\mu, \nu) := OT(\mu, \nu; || \cdot ||_p^p)^{1/p}$ [34, 35, 36]. **In this paper, we will set $p = 1, 2$ for the sake of convenience, respectively.**

## 4  Methodology

In this section, we introduce a new KGE method, termed *Optimal Transport Knowledge Graph Embeddings* (**OTKGE**) for multi-modal knowledge graphs. First, we present the definition of the multi-modal KGE problem.

**Problem Setup.**  Given a multi-modal knowledge graph $G = (\mathcal{E}, \mathcal{R}, \mathcal{T})$, where $\mathcal{E}$ is the set of entities, $\mathcal{R}$ is the set of relations, and $\mathcal{T} = \{(h, r, t) | h, t \in \mathcal{E}, r \in \mathcal{R}\}$ is the set of KG triples. Here the entity $e$ is equipped with a **structural embedding $e_S$**, **linguistic embedding $e_I$** and **visual embedding $e_V$**. Specifically, the uni-modal KGE method only focuses on $e_S$ while multi-modal KGE can leverage multi-modal embeddings.

**Representations for Entities and Relations.**  An important procedure in the KGE is to perform semantic matching between entities. To this end, previous work mainly focuses on studying how to model the relation and embedding space for uni-modal KGE. In this paper, we take a step beyond to leverage the multi-modal knowledge for the representation of entities and relations.

Supposing we have obtained the structural embeddings (by uni-modal KGE or initialization) and multi-modal embeddings (by pretrained models) of the entity in KGs, then here our goal is to learn the unified embeddings. Due to the heterogeneity of multi-modal spaces, one can notice that there exists a gap (caused by factors such as distribution discrepancy and dimensional difference) between different multi-modal embeddings. To address such an issue, as shown in Figure 2, we propose to realize the multi-modal alignment by optimal transport, and then perform multi-modal fusion to obtain a unified representation.

**Multi-modal Alignment.**  Considering that multi-modal knowledge is the auxiliary information, we let multi-modal embeddings align with the structural embeddings. Here we take the modal alignment between linguistic space $\mathcal{I}$ and structural embeddings $\mathcal{S}$ as an example, and the process between visual space $\mathcal{V}$ and structural space $\mathcal{S}$ can be achieved similarly.

With the OT alignment strategy, we can model the process of modal alignment as a process of transferring different each dimension of modal embeddings into an alignment space (structural embedding space). Specifically, it can be divided into the following steps: 1) Estimate the distribution $\mu$ for linguistic embedding $\boldsymbol{E}_I$ and the distribution $\nu$ for structural embedding $\boldsymbol{E}_S$; 2) Find a transport coupling $\boldsymbol{T}$ from $\mu$ to $\nu$, where $T_{ij}$ gives the probability to transport from the $i$-th feature dimension

of $\boldsymbol{E}_I$ to the $j$-th dimension of $\boldsymbol{E}_S$; 3) Use $\boldsymbol{T}$ to calculate the barycenter, and then map $\boldsymbol{E}_I$ to the unified space $\mathcal{S}$ and form a new embedding $\hat{\boldsymbol{E}}_I$.

Following the aforementioned steps, we denote by $\boldsymbol{u} = \{u_i\}_{i=1}^n \in \Delta_n$ and $\boldsymbol{v} = \{v_j\}_{j=1}^m \in \Delta_m$ for the probabilistic simplexes of linguistic and structural embeddings, respectively. Let $\mu = \sum_{i=1}^n u_i \delta_{\boldsymbol{E}_I^i}$, $\nu = \sum_{j=1}^m v_j \delta_{\boldsymbol{E}_S^j}$ represent discrete distributions of linguistic and structural embeddings, respectively, where $\delta$ is the Dirac function. For the sake of convenience, we set $u_i = 1/m, v_i = 1/n$, where $m, n$ are the dimensionality of the embeddings in $E_I$ and $E_S$, respectively. Then we denote the joint distribution $\prod(\mu, \nu) : \prod(\mu, \nu) = \{\boldsymbol{T} \in \mathbb{R}_+^{n \times m} | \boldsymbol{T} \mathbf{1}_m = \boldsymbol{u}, \boldsymbol{T}^T \mathbf{1}_n = \boldsymbol{v}\}$, where $\mathbf{1}$ denotes an all-one vector. Afterward, we compute the OT coupling $\boldsymbol{T}$ between the measures $\mu$ and $\nu$, which can be formulated as follows:

$$\mathcal{W}_p^p(\mu, \nu) = \min_{\boldsymbol{T} \in \prod(\mu, \nu)} \sum_{i=1}^n \sum_{j=1}^m \boldsymbol{T}_{ij} \boldsymbol{C}_{ij} \tag{1}$$

where $\boldsymbol{C}_{ij}$ is the cost function that evaluates the distance between $\boldsymbol{E}_I^i$ and $\boldsymbol{E}_S^j$. Moreover, $\mathcal{W}(\mu, \nu)$ can measure the distribution discrepancy between $\boldsymbol{E}_I$ and $\boldsymbol{E}_S$.

**Multi-modal Fusion.** After obtaining the transport matrix $\boldsymbol{T}$, the linguistic embedding $\boldsymbol{E}_I$ can be transformed into target-aligned embedding $\hat{\boldsymbol{E}}_I$ with the barycenter-based strategies [34, 35, 36]:

$$\hat{\boldsymbol{E}}_I = \text{diag}(1/\boldsymbol{v})(\boldsymbol{T}^T + \Delta_T)\boldsymbol{E}_I \tag{2}$$

Once this has been done, the transported linguistic embedding $\hat{\boldsymbol{E}}_I$ is in the same space as structural embedding $\boldsymbol{E}_S$, where $\Delta_T$ is an adjustable transport parameter [37]. Subsequently, we can obtain the aligned visual embedding $\hat{\boldsymbol{E}}_V$ in a similar way. Then the next step is to fuse the aligned multi-modal embeddings ($\hat{\boldsymbol{E}}_I$ and $\hat{\boldsymbol{E}}_V$) with the structural embedding $\boldsymbol{E}_S$ utilizing the strategy inspired by [38]:

$$\boldsymbol{E}_F = \min_{\boldsymbol{E}} \frac{1}{3} \sum_{i \in \{I, V, S\}} \lambda_i \mathcal{W}(\hat{\boldsymbol{E}}_i, \boldsymbol{E}) \tag{3}$$

where $\boldsymbol{E}_F$ is the unified representation; $\mathcal{W}(\boldsymbol{E}_i, E)$ is the $\mathcal{W}_p^p$ distance between the distribution of $E_i$ and that of $E$; $\lambda_i$ represents the weight. The whole procedure is summarized in Algorithm 1.

---

**Algorithm 1:** Multi-modal representations fusion.

---

**Input:** Distributions $\mu_k$ supported by multi-modal representations $\boldsymbol{E}_k$, and structural distribution $\nu$ supported by the structural embeddings $\boldsymbol{E}_S$.
**Output:** Unified representation $\boldsymbol{E}_F$

**Initialize** The size of input representation $n_k$, the size of fused representation $m$, the adjustable transport parameter $\Delta_T$.
$\nu = \mathbf{1}_m/m$
**foreach** $k \in \{I, V, S\}$ **do**
$\quad \mu_k = \mathbf{1}_{n_k}/n_k$
$\quad \boldsymbol{C}_S[i, j] = \|\boldsymbol{E}_k[i] - \boldsymbol{E}_S[j]\|_2^2, \forall i \in [n_k], j \in [m] \quad$ # $[m]$ means $1, \cdots, m$
$\quad \boldsymbol{T}_k = OT(\mu_k, \nu, \boldsymbol{C}_S)$
$\quad \hat{\boldsymbol{E}}_k = diag(1/\boldsymbol{v})(\boldsymbol{T}_k^T + \Delta_T)\boldsymbol{E}_k$
$\boldsymbol{E}_F = \min_{\boldsymbol{E}} \frac{1}{3} \sum_{i \in \{I, V, S\}} \lambda_i \mathcal{W}(\hat{\boldsymbol{E}}_i, \boldsymbol{E})$

---

**Relational Transformations.** On top of the aforementioned process, we have obtained unified representations of entities. Subsequently, we transform the head entity $h$ in the triple $(h, r, t)$ with the assistance of $r$:

$$\boldsymbol{h}_{relation} = \boldsymbol{r} \otimes \boldsymbol{h} \tag{4}$$

where $\boldsymbol{h}$ is the embedding of the head entity $h$, $\boldsymbol{r}$ is the embedding of the relation $r$, $\boldsymbol{h}_{relation}$ is the embedding of the transformed head entity. Here $\otimes$ is the transformations imposed to $h$ by $r$, which can refer to some uni-modal KGE mothods, *i.e.,* TransE and RotatE.

**Score Function and Loss.** Having obtained the transformed head entity, the score function formulates as follows:

$$f(h, r, t) = d(\boldsymbol{h}_{relation}, \boldsymbol{t}) \tag{5}$$

where $\boldsymbol{t}$ is the embedding of the tail entity $t$, and $d(\cdot, \cdot)$ represents a distance metric (*e.g.,* inner product or Euclidean distance). Afterward, to optimize the parameters, we train the model by minimizing the following loss:

$$L = \sum_{(h,r,t) \in \Omega \cup \Omega^-} log(1 + \exp(-Y_{label} f(h, r, t))) \tag{6}$$

where $Y_{label} \in \{-1, 1\}$ denotes the label of the triple $(h, r, t)$. Here suppose $\Omega$ denotes the set of observed triples, then let $\Omega^- = \mathcal{E} \times \mathcal{R} \times \mathcal{E} - \Omega$ be the set of unobserved triples. In the procedure of training, we adopt the negative sampling strategies (*e.g.,* uniform sampling or Bernoulli sampling [10]). Noticing that the multi-modal information is already pre-trained in advance, here we update the structural embedding in the training. This is also consistent with our original intention of using multi-modal information to assist KGE.

## 5 Theoretical Analysis

In this section, we will show that the OT-fusion strategy preserves consistency and comprehensiveness from a theoretical perspective.

First, we introduce the target error [39, 40] as an indicator to measure the distribution relationship between the target unified space and the source space. Here we first give some necessary definitions. Suppose $\mathcal{X}$, $\mathcal{Z}$ and $\mathcal{Y}$ are sets of entities, learned representations, and labels of triples, respectively. Notice that both structural embeddings and multi-modal embeddings of the entity are always assumed to obey the same true score function $f^* : \mathcal{Z} \to \mathcal{Y}$ in multi-modal KGE problems [18, 16, 19, 15, 20]. Considering the true scoring function $f^*$ is unknown, one usually chooses a predictor function $f$ from a hypothesis class $\mathcal{F}$ for substitution, $\forall f \in \mathcal{F}, f : \mathcal{Z} \to \mathcal{Y}$. To simplify the notations, here we take the head entity embedding $z \in \mathcal{Z}$ as the example to analyze while fixing the relation and the tail entity. The conclusion for the tail entity can be analyzed similarly.

Afterward, noticing that there exists a approximation error between the hypothesis $f$ and the true score function $f^*$ under the distribution $\mu$, we measure it here with $\epsilon_\mu(f, f^*) = \mathbb{E}_{z \in \mu}[|f(z) - f^*(z)|]$, which is termed target error. For simplicity of notation, we use the shorthand $\epsilon_\mu(f) = \epsilon_\mu(f, f^*)$. Moreover, we utilize the 1-Wasserstein distance $\mathcal{W}_1(\cdot, \cdot)$ to relate the source multi-modal distribution and target distribution. Above all, the target error in the OT fusion can be derived from the following theorem:

**Theorem 1 (Consistency and Comprehensiveness of the OT Fusion )** *We denote by $\mu_S$, $\mu_I$, $\mu_V$ and $\mu_F$ for the distributions of structural, linguistic, visual, and fused unified embeddings. Assume the hypotheses $f, f^* \in \mathcal{F}$ are all $K$-Lipschitz continuous for some $K$. Then the following statements hold for every hypothesis $f, f^* \in \mathcal{F}$:*

$$\max_{i \in \{I,V,S\}} \{\epsilon_i(f)\} \le \epsilon_F(f) + 2K \max_{i \in \{I,V,S\}} \{\mathcal{W}_1(\mu_i, \mu_F)\} \qquad \textbf{(Consistency)}$$

$$\epsilon_F(f) \le \min_{i \in \{I,V,S\}} \{\epsilon_i(f) + \mathcal{W}_1(\mu_i, \mu_F)\} \qquad \textbf{(Comprehensiveness)}$$

*where $\mathcal{W}_1(\cdot, \cdot)$ is the 1-Wasserstein distance, $\epsilon_*(f)$ is the target error at the corresponding embedding space $*$.*

The proof can refer to Appendix B.

**Remark 1 (For Consistency)** *In the RHS of the inequality, $\epsilon_F(f)$ is the target error of the unified embedding, which can be optimized implicitly by our training process; $\max_{i \in \{I,V,S\}} \{\mathcal{W}_1(\mu_i, \mu_F)\}$ is the maximum of the 1-Wasserstein distance between the individual modality and the unified space, which can be reduced by minimizing the alignment error in Eq.3. Meanwhile, the LHS represents the worst uni-modal performance. In this sense, the theorem above suggests that **improving the performance on the unified space can also improve the performance on individual modals**. Hence, the learned embeddings are ensured to be consistent with modal-specific information.*

**Remark 2 (For Comprehensiveness)** *Similar to the analysis of consistency, the second inequality in the theorem above shows that by minimizing the alignment error as in Eq.3, it is possible to obtain a reasonably unified embedding such that **the post-fusion error can be smaller than any involved uni-modal error**. This suggests that the OT-fusion strategy can integrate complementary information across different modalities efficiently.*

## 6 Experiments

### 6.1 Experimental Setup and Results

Table 1: Statistics of the datasets used in this paper. ($\text{Num}_e$ represents the number of entities and $\text{Num}_r$ represents the number of relations.)

| Dataset | $\text{Num}_e$ | $\text{Num}_r$ | Training | Validation | Test | Modality | Scale |
|---|---|---|---|---|---|---|---|
| WN9-IMG | 7k | 9 | 12k | 1k | 1k | Multi-modal | Small |
| FB-IMG | 11k | 1231 | 285k | 29k | 34k | Multi-modal | Large |

**Dataset.** In terms of the link prediction task, we conduct the experiments and evaluate OTKGE with two standard competition benchmarks as shown in Table 1. There includes multi-modal datasets: WN9-IMG [41] and FB-IMG [19]. M9-IMG dataset is derived from the subset of WN18 [9], which embraces structural knowledge as triples, and multi-modal knowledge including textual messages and visual images. FB-IMG dataset is derived from the subset of FB15K [9], which includes structural knowledge consisting of triples extracted from Freebase [42], and multi-modal knowledge embracing textual messages and visual images.

**Evaluation metrics.** To evaluate our model, we take entities in the knowledge replacing masked entities in the triples, then we rank all candidate triples through the score function. The evaluation metrics include the hit rate and the mean reciprocal rank. Specifically, we use H@n to represent the hit radio with cut-off values $n = 1, 3, 10$ while we use MRR to represent the mean reciprocal rank in the experiments.

**Baselines.** We compare our method to some representative models, *e.g.*, multi-modal KGE methods including IKRL [16], TBKGE [19], TransAE [15] and MMKRL [20]; uni-modal KGE methods including TransE [9], DistMult [25] ComplEx [11], and RotatE [12].

**Implementation Details.** In the course of the experiment, we implement OTKGE[2] with PyTorch and conduct experiments with a single GPU. To be fair, notice that the true triples will get top-rank positions when we conduct the evaluation. In this way, we adopt the filtered setting [9] to make sure that all true triples will be filtered out. Moreover, we utilize the grid search to find the hyper-parameters and we choose the best models from the validation set by using early stopping. Specifically, the embedding size $k$ is searched in $\{100, 200, 400, 500\}$ and the learning rate is searched in $\{0.001, 0.005, 0.01, 0.05, 0.1\}$.

- The structured embeddings are produced from triples in knowledge graphs, without any external multi-modal sources. To be specific, uni-modal KGE methods such as TransE [9] and ComplEx [11] can be used to learn structured embeddings.

- The linguistic embeddings of entities are learned by adopting the word2vec [43] technique. For instance, we learn the linguistic embeddings of FB-IMG dataset by pre-trained word2vec while we use GloVe [44] for the WN9-IMG dataset.

- The visual embeddings of entities are learned by pre-trained VGG [45] models. To be specific, visual embeddings are learned by adopting the VGG-m-128CNN [46] model in FB-IMG datasets. As for the WN9-IMG dataset, we take the VGG19 [45] model to learn visual embeddings.

---

[2]https://github.com/Lion-ZS/OTKGE

Table 2: Link prediction results on the multi-modal datasets (WN9-IMG and FB-IMG). If the indicator Is_M of a model is $\sqrt{}$, it means the model is a multi-modal KGE method; otherwise, it is a uni-modal KGE method. OTKGE$_1$ represents the version with 1-Wasserstein distance while OTKGE$_2$ represents the version with 2-Wasserstein distance. The best results are in **bold**, and the second-best results are underlined.

| Model | Is_M | FB-IMG | | | | WN9-IMG | | | |
|---|---|---|---|---|---|---|---|---|---|
| | | MRR | H@1 | H@3 | H@10 | MRR | H@1 | H@3 | H@10 |
| TransE | $\times$ | .712 | .618 | .781 | .859 | .865 | .765 | .816 | .871 |
| DistMult | $\times$ | .706 | .606 | .742 | .808 | .901 | .895 | .905 | .925 |
| ComplEx | $\times$ | .808 | .757 | .845 | .892 | .908 | .903 | .907 | .928 |
| RotatE | $\times$ | .794 | .744 | .827 | .883 | .910 | .901 | .914 | .926 |
| IKRL | $\sqrt{}$ | .742 | .691 | .785 | .844 | .898 | .894 | .908 | .922 |
| TransAE | $\sqrt{}$ | .755 | .698 | .794 | .857 | .901 | .900 | .912 | .928 |
| TBKGE | $\sqrt{}$ | .812 | .764 | .850 | .902 | .912 | .904 | .914 | .931 |
| MMKRL | $\sqrt{}$ | .827 | .783 | .857 | .906 | .913 | .905 | .917 | .932 |
| OTKGE$_1$ | $\sqrt{}$ | **.843** | **.799** | .873 | .914 | .920 | .909 | .925 | .940 |
| OTKGE$_2$ | $\sqrt{}$ | .842 | .798 | **.876** | **.916** | **.923** | **.911** | **.930** | **.947** |

**Results on multi-modal datasets.** To demonstrate the superiority of OTKGE, we compare it with other popular models on WN9-IMG and FB-IMG datasets as shown in Table 2. One can see that OTKGE achieves the best performance on both datasets. On one hand, the uni-modal KGE methods such as TransE can only learn the structural knowledge while the rich multi-modal information cannot be used effectively. Therefore, TransE does not perform as well as OTKGE, which demonstrates the advantage of multi-modal KGE. On the other hand, the multi-modal KGE methods such as IKRL use concat or mean to fuse information from multiple modalities, which will be constrained by spatial heterogeneity. Therefore, their experimental performance is weaker than OTKGE, which demonstrates the effectiveness of OT fusion.

## 6.2 Experiment Analysis

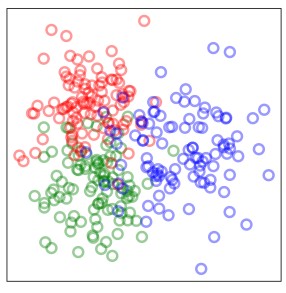 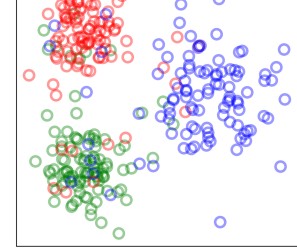 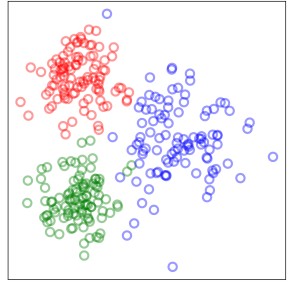

(a) Embeddings in TransE.     (b) Embeddings in IKRL.     (c) Embeddings in OTKGE.

Figure 3: The embedding comparison. Figure 3(a) represents structural embeddings learned by TransE, Figure 3(b) represents multi-modal unified embeddings learned by IKRL utilizing concat fusion and Figure 3(c) presents the multi-modal unified embeddings learned by OTKGE.

**The embedding comparison for OTKGE and other methods.** To show that OTKGE (here we adopt the version of 2-Wasserstein distance as an example) can learn a better-unified representation, we choose three kinds of semantic (people, place, and event) entities in the FB-IMG dataset and visualize the entity embeddings as shown in Figure 3. In these sub-figures, circles in blue (people), green (place), and red (event) represent three kinds of semantic embeddings, respectively. On one hand, one can see that using only structural knowledge in TransE can not separate these three kinds of embedding well, which demonstrates the necessity of leveraging multi-modal knowledge. On the other hand, there are still many entities that are not clearly separated in IKRL due to the heterogeneity of modal space. Different from this, OTKGE can effectively reduce the Wasserstein distance between

different distributions, facilitate the in-depth fusion of different modal knowledge, and thereby learn a better-unified representation.

**The ablation study of components in OTKGE.** As shown in Table 3, we conduct the experiments to study the role that different components play in OTKGE (here we adopt the version of 2-Wasserstein distance as an example). Specifically, we replace the fusion procedure via optimal transport by the mean and concat operations, which we denote as OTKGE w/ mean and OTKGE w/ concat; We only use the visual knowledge, which we denote as OTKGE w/o ling; we only use the linguistic knowledge, which we denote as OTKGE w/o visual. We can see that the performance of these ablation versions is reduced compared to the original version of OTKGE, which shows that the OT fusion plays an important role in OTKGE.

Table 3: Link prediction results on WN9-IMG and FB-IMG datasets.

| Model | FB-IMG | | | | WN9-IMG | | | |
|---|---|---|---|---|---|---|---|---|
| | MRR | H@1 | H@3 | H@10 | MRR | H@1 | H@3 | H@10 |
| OTKGE w/ mean | .828 | .789 | .857 | .899 | .913 | .909 | .915 | .921 |
| OTKGE w/ concat | .838 | .796 | .863 | .910 | .916 | .906 | .918 | .932 |
| OTKGE w/o ling | .835 | .791 | .867 | .908 | .914 | .907 | .914 | .929 |
| OTKGE w/o visual | .836 | .794 | .868 | .912 | .916 | .906 | .917 | .938 |

## 7 Conclusion

In this paper, we design a new KGE model named OTKGE for multi-modal knowledge graphs, where we formulate the fusion procedure by optimal transport theory. Previous work usually neglects the heterogeneity of distributions in the multi-modal fusion, which will do harm to the interaction of multi-modal knowledge. To tackle this problem, we propose to transfer the multi-modal information to a unified space by optimal transport, and fuse the multi-modal information with the Wasserstein barycenter. Theoretically, we prove that OTKGE is advantageous with its capability in learning multi-modal representation and the target error of the multi-modal embeddings and the unified embeddings can be bounded. Moreover, empirical experimental evaluations on multi-modal well-established datasets show that OTKGE can achieve overall state-of-the-art performance.

## 8 Acknowledgements

This work was supported in part by the National Key R&D Program of China under Grant 2018AAA0102000, in part by National Natural Science Foundation of China: U21B2038, 61931008, 61733007, U1936208, 6212200758 and 61976202, in part by the Fundamental Research Funds for the Central Universities, in part by Youth Innovation Promotion Association CAS, in part by the Strategic Priority Research Program of Chinese Academy of Sciences, Grant No. XDB28000000, in part by the China National Postdoctoral Program for Innovative Talents under Grant BX2021298, and in part by China Postdoctoral Science Foundation under Grant 2022M713101.

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
