# A  Algorithm for OTKGE

**Compared with the existing models.**  For the case of multi-modal KGEs, previous models such as IKRL learn the unified representation by concat or taking the mean of multi-modal representations. In this way, IKRL neglects the discrepancy of different multi-modal representations and it will harm the use of modal information. In contrast to this, OTKGE can measure distances between different multi-modal spaces by Wasserstein distance and consider the various distributional differences in these spaces. Intuitively, OTKGE can move different modal embeddings to a unified aligned space by an optimal transport plan while overcoming spatial heterogeneity by minimizing the Wasserstein distance between different distributions. It makes the process of multi-modal fusion more interpretational. In this sense, one can see that OTKGE shows strong advantages in multi-modal fusion.

# B  Proofs of Theorem 1

**Definition 1**  $f \in \mathcal{F}$ *is called K-Lipschitz continuous,* $\forall \boldsymbol{a}, \boldsymbol{b} \in \mathcal{D}$ *(where* $\mathcal{D} \in \mathbb{R}^n$*) if* $|f(\boldsymbol{a}) - f(\boldsymbol{b})| \leq Kd(\boldsymbol{a}, \boldsymbol{b})$.

Here are the proof for Theorem 1:

**Proof**  *First of all, we prove that* $|f - f'|$ *is* $2K$*-Lipschitz continuous given* $K$*-Lipschitz continuous hypotheses* $f, f' \in \mathcal{F}$. *we can derive the following formula with using the triangle inequality:*

$$
\begin{aligned}
|f(x) - f'(x)| &\leq |f(x) - f(y)| + |f(y) - f'(x)| \\
&\leq |f(x) - f(y)| + |f(y) - f'(y)| + |f'(y) - f'(x)|
\end{aligned}
\tag{7}
$$

*Suppose* $d(x, y)$ *represents a function to measure the distance between* $x$ *and* $y$*, for every* $x, y \in \mathcal{X}$*, then we have:*

$$
\begin{aligned}
\frac{|f(x) - f'(x)| - |f(y) - f'(y)|}{d(x, y)} &\leq \frac{|f(x) - f(y)| + |f'(x) - f'(y)|}{d(x, y)} \\
&\leq 2K
\end{aligned}
\tag{8}
$$

*In this step, we can find that for every hypothesis* $f, f'$*, given two distributions* $\mu_s$ *and* $\mu_t$ *(here* $\mu_s$ *is the multi-modal distribution while* $\mu_t$ *is the structural distribution), here we have*

$$
\begin{aligned}
\epsilon_t(f, f') - \epsilon_s(f, f') &= \mathbb{E}_{x \sim \mu_t}[|f(x) - f'(x)|] - \mathbb{E}_{x \sim \mu_s}[|f(x) - f'(x)|] \\
&\leq \sup_{\|f\|_L \leq 2K} \mathbb{E}_{\mu_t}[f(x)] - \mathbb{E}_{\mu_s}[f(x)] \\
&\leq 2K\mathcal{W}_1(\mu_s, \mu_t)
\end{aligned}
\tag{9}
$$

*where* $\mathcal{W}_{(\mu_s, \mu_t)}$ *is the 1-Wasserstein distance. Then we can derive the following formula:*

$$
\epsilon_t(f) \leq \epsilon_s(f) + 2K\mathcal{W}_1(\mu_s, \mu_t)
\tag{10}
$$

*By changing s,t, we have:*

$$
\begin{aligned}
\epsilon_I(f) &\leq \epsilon_F(f) + 2K\mathcal{W}_1(\mu_I, \mu_F) \\
\epsilon_V(f) &\leq \epsilon_F(f) + 2K\mathcal{W}_1(\mu_V, \mu_F) \\
\epsilon_S(f) &\leq \epsilon_F(f) + 2K\mathcal{W}_1(\mu_S, \mu_F) \\
\epsilon_F(f) &\leq \epsilon_I(f) + 2K\mathcal{W}_1(\mu_I, \mu_F) \\
\epsilon_F(f) &\leq \epsilon_V(f) + 2K\mathcal{W}_1(\mu_V, \mu_F) \\
\epsilon_F(f) &\leq \epsilon_S(f) + 2K\mathcal{W}_1(\mu_S, \mu_F)
\end{aligned}
$$

*Then the proof is completed.*