# OpenReview forum: "OTKGE: Multi-modal Knowledge Graph Embeddings via Optimal Transport"
_NeurIPS.cc/2022/Conference — NeurIPS 2022 Accept_

### Official Review · Reviewer_K5Tr · 2022-07-10

**Rating:** 7
**Confidence:** 4
**Soundness:** 2 fair
**Presentation:** 2 fair
**Contribution:** 2 fair

**Summary:**

This paper focuses on the multi-modal fusion problem. The authors propose a method to align the different modalities by reducing the Wasserstein distance between different modal distributions. Experimental results show that the proposed method achieves state-of-the-art performance.

**Questions:**

1 In Equation (4), the symbol E represents the structural embedding? The author should clarify.
2 In the experiment, the author should compare the proposed fusion method with other multi-modal fusion methods based on the aligned multi-modal embeddings.


**Limitations:**

Yes

**Strengths And Weaknesses:**

Strengths:
1 The paper is well written and easy to follow.
2 The motivation is clear.
3 The experiments and ablation studies well demonstrate the effectiveness of the proposed method.
Weaknesses：
1 The idea of aligning the different modalities is not novel.
2 Equation 4 is supposed to be further explained. What’s the insight behind the multi-modal fusion based on the Wasserstein barycenter?

---

> ### Author Response · Authors · 2022-08-02
> **Response to Reviewer K5Tr**
>
> We thank you for taking the time to carefully read our submission and providing such detailed suggestions. Our responses are as follows.
>
> __Q1:__ In Equation (4), is the symbol $E$ represent the structural embedding? Equation ($4$) is supposed to be further explained. What is the insight behind the multi-modal fusion based on the Wasserstein barycenter?
>
> __A1:__ The symbol $E$ represents the fusion embeddings, $E_1$ , $E_2$ and $E_3$ represent the structural embedding, the transported linguistic embedding, and transported visual embedding. In this sense, we have obtained structural embedding, visual embedding, and linguistic embedding, then Equation (4) shows that the fusion embedding is the closest point to them under the Wasserstein distance metric. In this way, the fusion embedding is the Wasserstein barycenter of structural, visual, and linguistic embeddings.
>
> __Q2:__ In the experiment, the author should compare the proposed fusion method with other multi-modal fusion methods based on the aligned multi-modal embeddings.
>
> __A2:__ Thank you for your considerable advice. We conduct the experiment to compare the proposed fusion method with other multi-modal fusion methods. We use the fusion method in [1] [2] and [3] to replace the OT fusion in OTKGE, and denote them as Model1, Model2, and Model3. We conduct the experiments in FBIMG datasets as shown in the following table. We can observe that the OT fusion can realize the best performance, the reason may lie in that the OT fusion in the OTKGE can better capture the embedded geometric information and overcome the heterogeneity of different embedded spaces.
>
> | Model    |MRR | Hit@1| Hit@3 |  Hit@10|
> | :-----------: |:---: |:-----------: | :---:|:---:|
> | Model1   |.595 |.578 | .606 | .670|
> |  Model2 |.594|.576 |.605|.668   |
> | Mode3|.596 |.581 |.609| .671|
> | OTKGE |.601|.583 |.615|.673   |
>
> [1] P. Lu, H. Li, W. Zhang, J. Wang, and X. Wang, “Co-attending free- form regions and detections with multi-modal multiplicative feature embedding for visual question answering,” in Proc. AAAI, 2018.
>
> [2] H. Ben-Younes, R. Cadene, N. Thome, and M. Cord, “BLOCK: Bilinear super diagonal fusion for visual question answering and visual relationship detection,” in Proc. AAAI, 2019.
>
> [3] J.-M. Pe ́rez-Ru ́a, V. Vielzeuf, S. Pateux, M. Baccouche, and F. Jurie, “MFAS: Multimodal fusion architecture search,” in Proc. CVPR, 2019.
>
> __Q3:__ The idea of aligning the different modalities is not novel.
>
> __A3:__ We are so sorry for the unclear presentation of the novelty. Our main novelty could be briefly rephrased as follows. To leverage a wealth of multi-modal knowledge and learn more realistic representations of real-world entities, previous work usually neglects the heterogeneity of distributions in the multi-modal fusion. It will do harm to the interaction of multi-modal knowledge. We propose to transfer the multi-modal information to a unified space by optimal transport and fuse the multi-modal information with the Wasserstein barycenter. It can effectively tackle the modal spatial heterogeneity by reducing the Wasserstein distance between different modal distributions.

---

### Official Review · Reviewer_2fCf · 2022-07-10

**Rating:** 6
**Confidence:** 3
**Soundness:** 3 good
**Presentation:** 3 good
**Contribution:** 3 good

**Summary:**

The paper studies the multi-modal knowledge graph embedding, which leverages textual, visual, and structure knowledge for learning more accurate representations of entities. The authors propose to utilize the optimal transport to transport different embedding distributions into a unified embedding space, then design a fusion strategy to obtain the final entity embeddings. The authors also provide theoretical analysis to support their claims.

**Questions:**

- I wonder why OTKGE outperforms other baselines in the uni-modal setting. Is that attributed to the design of relation r? However, we can see from Table 3 that OTKGE w/o trans has minimal performance degradation. It would be better if the authors could provide more detailed ablation studies on uni-model datasets.

**Limitations:**

Yes

**Strengths And Weaknesses:**

Strengths:
- The work is well motivated. Multi-modal data have heterogeneous spaces. How to fuse their representations is an interesting problem.
- It is novel to leverage the optimal transport to transfer different embedding distributions to a unified space.
- Theoretical analysis is provided to support the central claims.
- The paper is well written and easy to follow.


Weaknesses:
- Experimental results need more discussion and explanations.

---

> ### Author Response · Authors · 2022-08-02
> **Response to Reviewer 2fCf**
>
> Thank you so much for your acceptance as well as your constructive comments. The responses to your concerns are as follows.
>
> __Q1:__ Why OTKGE outperforms other baselines in the uni-modal setting? Is that attributed to the design of relation $r$?
>
> __A1:__ Relations play important roles in capturing the semantics in KGs and the ability to model relation $r$ can affect the performance of the model to a considerable extent.
> + (1) Recall that there are rich complex relations between entities in some uni-modal KGE datasets such as  FB15k-237, which embraces up to 237 kinds of relations. In this sense, it poses a challenge for learning semantic relations in knowledge graphs. To tackle this issue, the relation $r$ we designed servers the ability for OTKGE to model many key patterns, e.g., symmetry, anti-symmetry, inversion, composition, transitivity, hierarchy, intersection, and mutual exclusion patterns. In this way, it can handle these complex relationships above-mentioned in KGs and effectively capture latent semantics between entities.
> + (2) To study the role of relation $r$ played in uni-modal KGs, we conduct the experiments for uni-modal KG datasets. As shown in the following table, we denote the version of OTKGE which remove the transformation (Eq.(1)) as OTKGE w/o trans. One can observe that the performance of OTKGE w/o trans is reduced. Especially, the performance drop of OTKGE on FB15k-237 is more obvious, which shows the effectiveness of dealing with the complex relations with the design of the relation $r$.
>
> | Model   |Dataset |MRR | Hit@1| Hit@3 |  Hit@10|
> | :-----------: |:---: |:-----------: | :---:|:---:|:---:|
> |   OTKGE  | WN18RR |.495 |.449 | .508 | .571|
> | OTKGE w/o  trans|  WN18RR |.488|.441 |.495|.565   |
> | OTKGE|  FB15k-237  |.371 |.276 |.410| .560|
> | OTKGE w/o trans|  FB15k-237 |.357|.264 |.391|.542   |
> + (3) As for Table 3 in the paper, one can notice that OTKGE w/o trans has minimal performance degradation in multi-modal relations. The reason lies in that the multi-modal data provide rich auxiliary information, which yields gains for modeling complex relationships. Under such circumstances, we can observe that the design of $r$ has limited gain in model performance.

---

### Official Review · Reviewer_5SJG · 2022-07-11

**Rating:** 7
**Confidence:** 4
**Soundness:** 4 excellent
**Presentation:** 3 good
**Contribution:** 3 good

**Summary:**

This paper proposes optimal transport (OT) for the multi-modal fusion procedure of multi-modal KGE. It further provides theoretical analysis on target errors in OT fusion and generalization bounds for multi-modal KGE.

**Questions:**

1. Will the relationship in Eq. 12 always hold? If not, what are the possible cases that violate Eq.12?
2. In Line 245-246, the authors said that "The impact of the intrinsic complexity of function classes will be reduced when the sample size m is larger", I think this result is quite interesting. Could you provide some experimental results regarding this argument?


**Limitations:**

The limitations of this paper are properly discussed.

**Strengths And Weaknesses:**

Strengths
1. For previous multi-modal KGE methods, the embeddings from different modals are in various heterogeneous spaces, which is hard to avoid by using direct fusion. This paper proposes to mitigate the gap between heterogeneous spaces by using optimal transport (OT). The results of the ablation study clearly show the advantage of using OT for fusion over direction fusion counterparts like average or concatenation for fusion. Moreover, OTKGE can capture a wider range of Inference patterns than previous works.
2. Theoretical analysis of two important questions is given. Theorem 1 shows the target error can be bounded by the Wasserstein distances of different modals. Theorem 2 shows the generalization bound for the latent representation quality, and it further implies that multi-modal KGE can outperform uni-modal KGE analytically. In addition, the guarantee of Theorem 1 can not be easily achieved by other methods due to their fusion approaches. Theorem 2 justifies why multi-modal KGE is better. I think these theoretical results well distinguish the proposed methods from previous methods.
3. The experimental results are comprehensive, and the ablation study aligns with theoretical analysis.

Weaknesses
1. The organization of the theoretical analysis can be improved. It's better to split section 4 into two subsections. In addition, it would be better if the authors could state the relationships between Theorem 1 and Theorem 2 (if there are any). In addition, algorithm 1 and algorithm 2 can be moved into the main text instead of the appendix.

---

> ### Author Response · Authors · 2022-08-02
> **Response to Reviewer 5SJG**
>
> Thank you so much for your acceptance as well as your constructive comments. The responses to your concerns are as follows.
>
> __Q1:__ Will the relationship in Eq. $12$ always hold? If not, what are the possible cases that violate Eq.$12$?
>
> __A1:__ Generally speaking, the relationship in Eq.12 always holds empirically. However, there are also some cases that violate. For instance, if the information of the modality is inaccurate or noisy, the quality of entity representation learning will be affected or even reduced after adding this modality. In the circumstances, Eq.12 is not held. How to improve the quality of multimodal information or remove noise is also a direction we are interested in for future work.
>
> __Q2:__ Could you provide some experimental results regarding the argument that the impact of the intrinsic complexity of function classes will be reduced when the sample size $m$ is larger?
>
> __A2:__ Thank you for your useful advice. To be specific, the impact of the intrinsic complexity of function classes will be reduced when the sample size $m$ is larger, which means the performance difference between multi-modal KGE and uni-modal KGE will be also reduced. Based on this knowledge, we conduct the experiment with FBIMG datasets to demonstrate this argument. We denote FB%x as the dataset version that removes x percent of the data in the train datasets of FBIMG. Then the experimental result is shown as the following table. We can observe that as the number of samples increases, the performance difference between single-mode and multi-mode decreases.
> |Dataset|Modal |MRR | Hit@1| Hit@3 |  Hit@10|
> |:---: |:--:|:-----------: | :---:|:---:|:---:|
> | FB%30 |uni-modal  |.553 |.529 | .567 | .631|
> | FB%30 |multi-modal  |.575 |.553 | .579 | .655|
> | FB%20 | uni-modal |.582|.558 |.586|.657 |
> | FB%20 | multi-modal |.593|.567 |.595|.671 |
> | FB%10 | uni-modal |.596 |.578 |.602|.665|
> | FB%10| multi-modal |.601|.583 |.615|.673   |

---

### Public Comment · ~Haotong_Yang1 · 2022-12-13
**Is the code available?**

I like this work and wondering about some details. I find that in the checklist the authors say the code is included in the supplemental material, but I do not find it. Where can I find your code? Thank you for your reply!

---

> ### Public Comment · ~Zongsheng_Cao1 · 2022-12-13
> **Reply for some details**
>
> Thank you for your attention, and the code has been released at https://github.com/Lion-ZS/OTKGE.

---

### Meta-Review · Area_Chair_wTTW · 2022-08-30

**Recommendation:** Accept
**Confidence:** Certain

**Metareview:**

This paper presents a method to learn multi-modal knowledge graph embeddings. To integrate the embeddings from different modalities, which is a difficult task because of the heterogeneity across the different modalities, the paper presents an optimal transport based method to learn multi-modal embeddings.

The paper received positive reviews from all the reviewers. The authors submitted a rebuttal to answer the questions from the reviewers, and the reviewers seem to be satisfied.

Given the unanimously positive reviews and my own reading of the paper, I vote for the acceptance of the paper.

**Award:**

No

---

### Decision · Program_Chairs · 2022-09-14

Accept